# Factors Influencing Exclusive Breastfeeding Amongst Arab Mothers in Israel: Social, Cultural, and Occupational Aspects

**DOI:** 10.3390/healthcare13080852

**Published:** 2025-04-08

**Authors:** Khaled Awawdi, Orsan Yahya, Mohammad Sabbah, Sana Bisharat, Mahdi Tarabeih

**Affiliations:** 1Faculty of Health Sciences, Department of Nursing, Ramat Gan Academic College, 87 Pinhas Rotenberg Street, Ramat-Gan 5227500, Israel; awawdi.h@iac.ac.il (K.A.); hamodi_sabbah@iac.ac.il (M.S.); 2The Azrieli Faculty of Medicine, Bar-Ilan University, 8 Henrietta Szold Street, Safed 1311502, Israel; orsanya@clalit.org.il; 3Department of Family Medicine, Clalit Health Services, Afula 1710601, Israel; 4Rambam Health Care Campus, 8 HaAliya HaShniya Street, Haifa 3109601, Israel; 5The Holy Family Hospital Nazareth, Hagalil Street, Nazareth 1641105, Israel; sana@hfhosp.org; 6School of Nursing Sciences, The Academic College of Tel-Aviv-Yaffa, Rabenu Yeruham Street, 8401, Yaffo 6818211, Israel

**Keywords:** exclusive breastfeeding, Arab mothers, religiosity, maternal health, breastfeeding duration, cultural beliefs, employment status, socio-demographic factors, child nutrition, Israel

## Abstract

**Background:** Exclusive breastfeeding (EB), feeding an infant solely breast milk, has been recommended worldwide due to its health benefits for infants and mothers. However, EB rates remain low, due to several socio-demographic, occupational, and cultural factors. **Objectives:** Our aim was to investigate the factors associated with EB amongst Arab mothers living in Israel, i.e., age, education, religiosity, employment status, and the number of children which impact breastfeeding practices. **Methods:** Data were collected through surveys from Arab mothers of different religious and geographical backgrounds. **Results:** The findings revealed that older maternal age, higher levels of religiosity, and an increased number of children were positively associated with EB. Employment reduced EB. Muslim mothers were more likely to combine breastfeeding with formula; Christian and Druze mothers showed higher EB rates. Geographical disparities were also noted. **Conclusions:** This study highlights the complex interplay of socio-demographic and cultural factors that impact breastfeeding practices amongst Arab mothers in Israel, with significant implications for policy development and maternal support programs. We underscore the importance of incorporating religious and cultural considerations into breastfeeding promotion strategies. Public health initiatives should target support for working mothers by advocating extended maternity leave and breastfeeding-friendly workplace policies. Furthermore, community-based interventions should engage family members in supporting breastfeeding mothers.

## 1. Introduction

The practice of exclusive breastfeeding (EB) from birth to four months of age is common worldwide. Health organizations, such as the World Health Organization (WHO) and the Israeli Ministry of Health, recommend EB up to 6 months of age as the ideal goal, which will be followed by the introduction of solid foods and continued breastfeeding. Before six months, infants are typically not yet developmentally ready to handle solid foods, thus rendering breastfeeding the ideal source of nutrition at this stage. EB also provides natural antibodies which bolster the infant’s immune system.

The first two years of the baby’s life are critical, and proper nutrition is essential for proper growth and development [1]. Inadequate nutrition increases the risk of both short- and long-term morbidity and mortality, i.e., wasting (low weight for height), stunting (delayed growth), obesity, cognitive impairment, diminished future physical work capacity, and, in female infants, fertility problems [1,2,3]. Moreover, poor early nutrition can lead to the development of unhealthy eating habits that may persist throughout life [4]. According to data from the WHO and the World Bank, 6.8% of children under 5 years old suffer from wasting, and ~22% experience stunting. The majority of these cases occur in children from developing or third-world countries (mostly Africa and Asia). However, these conditions are also present in developed countries, with prevalence rates of stunting and wasting at 4% and 0.4%, respectively. Globally, a common nutritional problem is food insecurity, defined as the limited or uncertain availability of safe and adequate food or an unreliable ability to acquire food through socially acceptable means. In Israel, a developed country, ~21% of children suffer from food insecurity, with approximately half of these children experiencing severe food insecurity [5]. 

Childhood obesity, on the other hand, is a more prevalent phenomenon found in developed countries—7.6% vs. 3.4% in developing countries [6]. Among breastfeeding’s numerous advantages for both infants and mothers is the reduced risk of obesity as well as the development of an emotional bond/attachment between mother and infant. Moreover, there is a decreased risk of breast and ovarian cancer for the mother [7,8,9]. Studies have shown that breastfed infants have a 26% lower likelihood of suffering from obesity or overweight [10,11,12]. 

Research has also indicated that EB offers additional benefits, including a reduced risk of mortality in infants due to sudden infant death syndrome (SIDS). A meta-analysis conducted in the USA found that the risk of SIDS was reduced by 45% in infants who were breastfed (either exclusively or partially); in infants breastfed for at least two months, the risk decreased by 62%, and in infants who were breastfed for a certain amount of time, by 73% [13,14]. Meek and Noble found that EB may bestow a protective effect against the development of asthma, eczema, lower respiratory tract infections, and type 1 diabetes [15]. Despite the advantages associated with EB, a significant percentage of women worldwide do not breastfeed.

Only three developed countries (USA, Spain, and France) have reported that >80% of mothers have ever breastfed [16]. In 2020, according to the Center for Disease Control (CDC) data from the USA, the breastfeeding initiation rate was 83.1%, with significant differences observed across various socio-economic groups (90.1%, among women in the highest socio-economic versus 75.4% in the lowest socio-economic bracket). The breastfeeding initiation rate among university-educated women was 91.9%, compared to 72.5% who did not complete high school [17]. These data indicate that only 45.3% of infants received EB at 3 months; and 25.4% at 6 months of age [17]. According to the Global Breastfeeding Collective of the WHO and UNICEF, only 48% of infants under 6 months are EB [18]. To encourage breastfeeding, many hotlines for breastfeeding mothers have been set up internationally, i.e., the Australian Breastfeeding Association website, which enumerates the advantages of breastfeeding for both mothers and infants and offers help with related problems (Australian Breastfeeding Association website).

### 1.1. Breastfeeding Practice in Israel

A national health and nutrition survey conducted in Israel between 2019 and 2020 [19] found that the percentage of women practicing EB for a duration of 4 to <6 months was 28.8% [20]. In another survey conducted in Israel, it was reported that the initiation rate of breastfeeding was ~90%, for all women who began EB in hospital. However, the rates of EB significantly declined during the postpartum period (during maternity leave). The survey indicated that the primary reason for discontinuing breastfeeding was difficulty breastfeeding, i.e., physiological issues and problems with milk supply [19]. 

Various factors influence the duration of EB, i.e., demographic and biological, perceptions and beliefs, social, and hospital-related. Demographic factors include age (older women are expected to breastfeed more than younger women); education level (the higher the education, the greater the likelihood of EB); and income level (as income increases, the likelihood of breastfeeding also increases [21,22]. Biological factors primarily relate to the ability to produce milk or a sufficient milk quantity. Moreover, research has shown that mothers who are obese at the time of pregnancy exhibit a lower likelihood of breastfeeding. Factors related to the mother’s perceptions and beliefs include feelings of self-efficacy (the higher the self-efficacy, the greater the likelihood of breastfeeding) [22,23].

Social factors include the mother’s employment status, duration of maternity leave, and support from her partner. Hospital-related factors include policies (such as rooming-in and early initiation of breastfeeding) [24,25]. It is essential to raise awareness regarding the long-term importance of breastfeeding during the mother’s visits to well-baby clinics. The WHO, the American Academy of Pediatrics (AAP), and the Israeli Ministry of Health recommend EB for the first six months following the birth of the infant [25,26,27,28,29]. Breastfeeding patterns among Arab mothers in Israel, as demonstrated in several national health surveys and academic studies, have exhibited variations in breastfeeding initiation and exclusivity rates among Muslim, Christian, and Druze women [30,31,32].

### 1.2. Religious Significance Attributed to Breastfeeding

In the Israeli Arab community (Islam, Christianity, and Druze), breastfeeding holds a significant symbolic value due to religious reasons, hence, we focused on three groups-a non-conservative, conservative, and a religious group. A majority of Arab women do not participate in the labor force, remaining at home to raise their children. Consequently, breastfeeding is perceived as a fundamental and important issue, similarly to global trends, as a result of its health benefits for both infants and mothers. However, also within this community, cultural, social, and economic factors affect attitudes and practices related to breastfeeding.

In terms of traditional and religious significance, breastfeeding is regarded in Islam as a religious duty. The Quran specifies the duration of breastfeeding in the verse, “Mothers may breastfeed their children for two complete years for whoever wishes to complete the nursing” [33]. Breastfeeding is considered a religious obligation of the mother towards the child; however, the child may be transitioned to artificial feeding, if necessary for the child’s welfare. If a mother is unable to breastfeed her child, another woman (a wet nurse) is appointed to the role and establishes a bond of “milk kinship” between the infant, wet nurse, and family, a relationship that is recognized and precisely defined by Islamic religious law, giving breastfeeding a spiritual value [34,35].

In Christianity, breastfeeding is viewed as a maternal duty representing care, love, and nourishment and a symbol of the spiritual relationship between the believer and the Church. Christian religious imagery often portrays the Virgin Mary breastfeeding Jesus In the Catholic tradition, Mary is often portrayed as the ideal mother. Images of Mary breastfeeding the infant Jesus are perceived as a representation of how she provides life to her son, both physically and spiritually [36]. In the Druze community, breastfeeding is perceived as an essential part of raising a child, and is considered an integral part of a mother’s responsibility towards her child, signifying closeness and love [30].

The correlation between religiosity and breastfeeding is well-supported in the literature. Dorri et al. [31] found that religious values and family support significantly influenced Arab immigrant mothers in their breastfeeding initiation and exclusivity. Similar patterns have been observed in other countries such as Pakistan, where Arif et al. [37] found that religious affiliation correlated with longer breastfeeding duration. Religious teachings, cultural norms, family support, and lower workforce participation all contribute to higher exclusive breastfeeding rates amongst religious mothers. This trend is not unique to Arab mothers in Israel but, it has also been globally documented across various religious and ethnic groups. Previous research has highlighted the influence of religious and cultural beliefs on breastfeeding duration and exclusivity among Arab communities in Israel [26,32].

### 1.3. Family and Social Factors

In many Arab families, especially in religious communities, there is strong support for breastfeeding from extended family members, such as grandmothers and aunts, who provide guidance on the benefits and techniques of breastfeeding. There is often encouragement to breastfeed for extended periods, and in some cases, even beyond the first year of the child’s life. Community norms within religious societies may discourage formula feeding, associating breastfeeding with traditional maternal roles [32]. However, the level of familial support and influence may vary across families. Furthermore, various social factors also impact breastfeeding, i.e., education, age, socio-economic status, ethnic background, health, and cultural beliefs. Socio-economic status also plays a pivotal role. For example, in the United States and the United Kingdom, non-white ethnic groups, particularly, those of Caribbean descent, tend to have lower breastfeeding rates. Cultural beliefs, in particular, exert a significant influence on the likelihood of breastfeeding [31]. 

Because breastfeeding in the Israeli Arab community is regarded as a key element with significant health, cultural, and religious value, research has reported a steady increase in breastfeeding rates among Israeli Arab women, with studies showing that in the initial months following childbirth, breastfeeding rates rise to between 85% and 90%. Among Muslim women, there is a tendency to breastfeed for extended periods of time, sometimes up to a year or beyond [30,31,36,37].

Breastfeeding rates among Christian Arab women are lower than among Muslim women, yet, above the national average. Research has shown that the breastfeeding rate among Christian women ranges from 70% to 80% during the initial months postpartum, with a gradual decline thereafter, similar to trends observed in other populations [34,37]. Furthermore, among Druze women, breastfeeding rates are also high, reaching ~85% to 90% in the months following childbirth. Similarly to their Muslim counterparts, Druze women perceive breastfeeding as having significant cultural and communal value. Nonetheless, there is a decline in breastfeeding rates associated with the adoption of modern lifestyles and economic pressures, i.e., the necessity to return to work [30,31,37,38]. Studies have demonstrated that employment status, education level, and socio-economic status significantly impact breastfeeding duration, with these working mothers facing greater challenges in maintaining exclusive breastfeeding [37,39]. In addition, regional disparities were observed in breastfeeding practices. Research has shown that Arab mothers in northern Israel exhibit higher rates of exclusive breastfeeding compared to those in central and southern regions, reflecting variations in access to healthcare resources and community support [38].

Overall, breastfeeding rates are relatively high across all groups, with the highest rates observed among Druze women, followed by Muslim women, and lastly, Christian women. There is a general trend of declining breastfeeding rates over time, correlated with the increasing complexity of modern life. The relationship between poverty, women’s occupations, and EB has been extensively studied. Health-related initiatives, such as the WHO Code and the Baby-Friendly Hospital Initiative have addressed these factors [35,38]. Analyzing economic gradient data can further clarify the impact of education and occupation on breastfeeding trends [40].

Therefore, the objective of this study was to examine the impact of socio-demographic, occupational, and religious characteristics on the rates of EB among Israeli Arab mothers (Muslim, Christian, and Druze). We investigated how factors such as age, education, marital status, income level, and profession affect a mother’s decision to EB. Additionally, we explored the relationship between the mothers’ level of religiosity and EB rates, endeavoring to understand the contributions of these factors to the duration and/or continuation of EB and related public health policies.

The research hypotheses were:

**H1:** 
*As the mother’s age increases, the likelihood of EB and its duration increases.*


**H2:** 
*The more previous births (i.e., children), the likelihood of EB and its duration increases.*


**H3:** 
*There are differences in the rate of EB and its duration based on ethnic group affiliation and geographical region.*


**H4:** 
*As the mother’s education level increases, the likelihood of EB and its duration increases.*


**H5:** 
*As the mother’s religiosity level increases, the likelihood of EB and its duration increases.*


**H6:** 
*There are differences in the rates of EB and its duration based on the type of breastfeeding instructional training/courses the mother had received.*


**H7:** 
*Working reduced the likelihood of both EB and its duration, further decreasing as the workload (scope) increases.*


**H8:** 
*An association exists between the mother’s education level and the factors which foster breastfeeding and those that lead to its cessation.*


While hypotheses H1–H4 address aspects that have been studied to some extent in the existing literature, they remain integral to our study because they provide validation of previously observed trends among the underrepresented population of Arab mothers in Israel. Furthermore, they allow for a comparative analysis to assess whether these well-documented relationships hold within our specific sample or show unique patterns affected by religiosity, employment status, or regional factors, thus serving as a foundation for understanding the broader interplay of factors crucial for interpreting findings from the newer hypotheses (H5–H8) and ensuring a comprehensive analysis.

While previous studies have analyzed general breastfeeding trends, there is a lack of comparative data on Muslim, Christian, and Druze mothers, especially in the context of exclusive breastfeeding. There is also insufficient research on regional disparities in breastfeeding practices within Israel, particularly in the north versus central and southern regions.

By addressing these gaps, this study provides a comprehensive analysis of the interplay between religiosity, socio-economic factors, and breastfeeding habits among Arab mothers in Israel.

Our findings offer critical insights for public health policies, especially for designing culturally sensitive breastfeeding promotion programs. The study also contributes to the broader discussion on breastfeeding disparities in minority populations within developed countries.

## 2. Methods

### 2.1. Ethical Approval

The study was conducted according to the guidelines of the Declaration of Helsinki and approved by the Institutional Review Board of the Ramat Gan Academic College, approval code #2023-1010, approval date: 20 July 2023. Part I of Appendix A: Informed Consent Form for Online Studies, explains to each participant who answered the questionnaire that “by clicking the ‘I agree’ button, you express your consent to participate in the study”. Thereby, every respondent to the questionnaire gave her informed consent (Appendix A: Informed Consent Form).

### 2.2. Procedure

The target population for this study were Arab women (Muslim, Christian, and Druze) from the age of 18, who had given birth within the past year and breastfed their infants for at least four months postpartum. The research tool was a questionnaire composed by a team of family health nurses from the Israeli Ministry of Health and validated by experts in the field of breastfeeding—three family health nurses and two family health specialists. The questionnaire was originally written in Hebrew and translated into Arabic, then translated back into Hebrew and reviewed by translators who specialize in questionnaire language to ensure that it was easy to understand. The initial sample included questionnaires filled in by 300 participants; however, 26 questionnaires were excluded due to incomplete responses. Consequently, the final sample comprised responses from 274 participants who met the predetermined inclusion criteria. These participants were drawn from three different regions of Israel (north, central, and south) and represented a variety of age groups and socioeconomic backgrounds.

The sampling method utilized a snowball approach, wherein participants assisted the researchers in recruiting other potential participants through their personal networks. Additionally, the questionnaire link was made available online, and physical copies were distributed to well-baby clinic (Tipat Chalav) nurses and given to breastfeeding women. The exclusion criteria were a minimum age of 18, having given birth within the year, and having breastfed the infant for at least four months. The researchers had no prior relationship or familiarity with any of the participants, all of whom received detailed information about the study. Anonymity and confidentiality were assured, and participants were informed that their involvement would not provide any personal benefits or advantages but would contribute to the advancement of general knowledge in this area of research.

There were no known risks associated with participation in the study; however, as with any online activity, there was a potential risk of privacy breaches. Researchers made every effort to mitigate this risk by ensuring that the questionnaires were completed anonymously, with personal details used solely for research purposes. Participants were informed that additional information could be obtained from the Ethics Committee via an inquiry form at the conclusion of the study. The research team could also be contacted through the email address provided in the questionnaire. Participants retained the right to discontinue the survey at any stage. Informed consent was explicitly obtained through a designated question at the beginning of the survey. Only after providing their informed consent could they access the questionnaire link and proceed to complete the survey.

### 2.3. Participants

The sample consisted of 274 married Israeli Arab women between the ages of 21–48 (M = 31.05, SD = 5.21). Of these, 186 (67.9%) respondents were Muslim, 40 (14.6%) were Christian, and 48 (17.5%) were Druze. Their religiosity level ranged from 1 to 5 (M = 4.55 and SD = 0.74, Cronbach’s Alpha Coefficient, α = 0.83), indicating a considerably religious participant group. The sample was predominantly drawn from Arab communities in northern Israel (80.3%), with a lower representation from the central (6.9%) and southern (12.85%) regions. The snowball sampling method was used due to accessibility constraints, which may have influenced the geographic distribution of participants. To mitigate this bias, efforts were made to distribute the surveys through well-baby clinics (Tipat Chalav) and online platforms in order to reach a broader sample of Arab mothers (Table 1).

### 2.4. Measures

The research questionnaire was based on a survey from the local National Center for Disease Control (ICDC) reporting on baby nutrition. The questionnaire was then approved by five experts in the field, and translated into Arabic (the native language of the participants). This was carried out to ensure maximum accessibility and more accurate and representative responses. Prior to distributing the survey, a pilot questionnaire which had been answered by 40 individuals was collected in order to assess the questionnaire. Subsequently, minor changes were made.

The survey comprised three sections: (1) socio-demographical information (i.e., age, marital status, educational level); (2) baby nutrition-related questions (i.e., type/method of feeding, duration of breastfeeding, instructional training regarding breastfeeding); and (3) the impact of factors that either foster or halt breastfeeding habits (Appendix A and Appendix B).

#### Measurement of Religiosity

The level of religious belief was measured using a 5-point Likert scale, where participants rated their agreement with statements regarding their religious practices and beliefs:

(1) Strongly disagree, (2) Disagree, (3) Neutral, (4) Agree, (5) Strongly agree.

Three specific items were assessed: “I believe in religious values”, “I act according to traditional religious values”, and “I observe the commandments of my faith”.

The total religiosity score was calculated as the mean score across these items (Cronbach’s alpha = 0.83), ensuring internal reliability.

### 2.5. Data Analysis

Statistical Software and Preprocessing: Data were analyzed using SPSS v28. Before hypothesis testing, we conducted data cleaning and normality checks using Shapiro–Wilk tests for continuous variables. Descriptive statistics (means, standard deviations, and frequencies) were used to summarize the sample characteristics.

Hypothesis Testing and Statistical Methods: One-way ANOVA was used to compare mean differences between multiple groups (e.g., age groups and breastfeeding duration). Chi-square (χ^2^) tests were employed to examine associations between categorical variables, such as employment status and breastfeeding type. Finally, Pearson’s correlation tests were conducted to assess the strength and direction of relationships between continuous variables, such as maternal age and breastfeeding duration.

Post Hoc Analysis for ANOVA: When ANOVA results indicated statistically significant group differences (*p* < 0.05), Tukey’s Honestly Significant Difference (HSD) post hoc tests were conducted to identify specific group differences. For non-parametric comparisons, Bonferroni-adjusted pairwise comparisons were used to control for Type I errors.

Effect Size Measures: Eta-squared (η^2^) was calculated for ANOVA to determine the magnitude of the effect. Cramer’s V was reported for chi-square tests to assess the strength of associations.

Statistical Significance and Confidence Intervals: A significance threshold of *p* < 0.05 was used for all analyses. Confidence intervals (CIs) of 95% were provided to indicate the precision of estimates.

The analysis of variables was carried out thusly: breastfeeding methods (exclusive breastfeeding, combined breastfeeding, or formula feeding) were analyzed using chi-square tests to assess differences between groups.

Breastfeeding duration (categorized as up to 1 month, 2–3 months, 4–6 months, and more than 6 months) was analyzed using one-way ANOVA to determine significant variations across different demographic and religious groups.

Religiosity and breastfeeding behavior were examined using Tukey’s HSD post hoc tests, following significant ANOVA results, to identify specific differences among levels of religiosity.

Employment status and breastfeeding were analyzed using logistic regression models to evaluate the impact of full-time, part-time, and unemployment on exclusive breastfeeding rates.

## 3. Results

### 3.1. Participant Characteristics

Education-wise, more than half of the participants who responded to the survey were academics (58.4%), 9.9% did not finish high-school, 23.4% completed high-school, and 8.4% had learned in a tertiary education setting (Table 1).

Workwise, most were salaried part-/full-time employees (64.2%), 29.6% were unemployed, and the rest were self-employed (6.2%). Some participants did not work prior to the current birth (30.7%), whereas 29.6%, worked full time (100%), 24.1%, worked 75%, 12% worked 50%, and 3.6% worked 25%. When asked, 37.2% reported that they planned on returning to work after giving birth, 28.5% had already returned to work, 8.8% stated that they would not return, and 25.5% were uncertain. Moreover, when asked, the respondents declared that they planned to return to work when their baby reached the age of (1) 1 month (0.4%), (2) 2 months (0.4%), (3) 3 months (7.3%), (4) 4–6 months (15%), (5) 6 months and above (40.9%), and 36.1% stated that they would not return to work. Household-wise, 38.3% had had no children before their recent delivery, 19.3% had only one child prior to their recent delivery, 17.5% had two children, and 24.8% had three children. Typically, the participants housed 1 more individual in their home (2.2%), 2 individuals (8%), 3 individuals (23.4%), and >4 (66.4%) (Table 1).

### 3.2. Hypotheses Testing

In order to test hypothesis H1, the assessment of age differences between (1) the three different breastfeeding methods (i.e., complete reliance on baby food and/or supplements, EB, or integrated/combined EB and baby food) and (2) breastfeeding duration, one-way analyses of variance (one-way ANOVAs), were employed.

In assessing the association between age and breastfeeding methods (exclusive; N = 165, M = 31.73, SD = 4.76/combined; N = 66, M = 30.00, SD = 4.92/baby food; N = 43, M = 29.6, SD = 3.28), the results indicated a significant main effect: *F*(2, 271) = 3.60, *p* < 0.05. Hence, in order to gauge the source of these significant differences, Tukey’s HSD post hoc tests were performed, revealing only that those who preferred EB were slightly older than those who opted for combined breastfeeding (*p* < 0.05). No other significant differences were discovered between these groups.

In assessing the association between age and breastfeeding duration during the last 4 months (did not breastfeed at all; N = 19, M = 28.95, SD = 3.52/up to 1 month; N = 10, M = 27.20, SD = 4.61/up to 2 months; N = 6, M = 27.33, SD = 5.05/up to 3 months; N = 25, M = 32.20, SD = 8.73/up to 4 months; N = 20, M = 28.45, SD = 5.83/above 4 months; N = 194, M = 31.69, SD = 4.46), the results indicated a significant main effect: *F*(5, 268) = 4.40, *p* < 0.01. In order to gauge the source of these significant differences, Tukey’s HSD post hoc tests were performed, revealing that those who breastfed for >4 months were slightly older than those who did so up to 4 months (*p* < 0.05) and up to 1 month (*p* < 0.05). No other significant differences were discovered between these groups.

In order to test hypothesis H2, the assessment of the association between the number of children prior to the recent delivery and (1) the three different breastfeeding methods (i.e., only baby food, EB, integrated/combined breastfeeding) and (2) breastfeeding duration, chi-square tests were performed. Statistically significant differences were found among the different breastfeeding methods—χ^2^ (6, N = 274) = 41.15, *p* < 0.01, *r_c_* = 0.27. Results indicated that (1) those with no children before the current delivery preferred baby food or combined breastfeeding; (2) those with one child were divided between the three different breastfeeding methods (respectively, 28.6%, 29.5%, and 41.9%); All the others preferred EB in this order-first, those with three children (86.8%), second those with two children (62.5%), third those with one child (60.4%); statistically significant differences were also found amongst those with a number of children before the last birth with breastfeeding duration: (6, N = 274) = 45.55, *p* < 0.01, *r_c_* = 0.23. All the women in the sample preferred to breastfeed for more than 4 months, in this order -firstly, women with three children, 89.7%, preferred breastfeeding longer than 4 months; secondly, women with two children, 81.3%; thirdly, women with one child 71.7%; fourthly, women with no previous children 53.3% (Table 2 and Table 3).

In order to test hypothesis H3, the assessment of the association between ethnic affiliation (Muslim Arabs/Christian Arabs/Druze) or geographic region (the north, center, and south of the country) and (1) the three different breastfeeding methods (i.e., only baby food, EB, integrated/combined breastfeeding) and (2) breastfeeding duration, chi-square tests were performed.

Statistically significant differences were found among the different ethnic groups—χ^2^ (4, N = 274) = 31.93, *p* < 0.01, *r_c_* = 0.24. The results indicate that (1) Muslim women were divided between the three different breastfeeding methods (respectively, 20.4%, 29.6%, and 50%) and (2) Christian Arabs (67.5%) and Druze (93.8%) preferred EB (Table 4).

Statistically significant differences were found among the different ethnic groups—χ^2^ (10, N = 274) = 22.73, *p* < 0.01, *r_c_* = 0.20. The results indicate that (1) Muslim Arab women preferred to breastfeed for >4 months. Firstly, the Druze, 95.8%; secondly, the Christian Arabs, 75%; thirdly, the Muslim Arabs, 63.4% (Table 5).

Statistically significant differences were found among the different regions: χ^2^ (4, N = 274) = 14.70, *p* < 0.01, *r_c_* = 0.16. The results indicate that (1) those living in the north (63.2%) and south (60.2%) of the country preferred EB and (2) those living in the center were divided between the three different breastfeeding methods (respectively, 31.6%, 21.1%, and 47.4%). Statistically significant differences were found among the different regions and breastfeeding durations: χ^2^ (10, N = 274) = 25.36, *p* < 0.01, *r_c_* = 0.22. The results indicate that women from all regions preferred to breastfeed >4 months; firstly, women from the north, 75.9%; secondly, women from the center, 52.6%; thirdly, women from the South, 48.6% (Table 6 and Table 7).

There are several possible socioeconomic and cultural explanations for why Christian and Druze mothers show higher exclusive breastfeeding rates than Muslim mothers. These relate to employment and workforce participation—Muslim Arab women in Israel have higher workforce participation rates compared to Druze women, many of whom tend to stay at home longer post birth due to traditional family structures. This might allow Druze mothers to breastfeed for longer durations.

In order to test hypothesis H4, the assessment of the association between educational level (non-academic/academic) and (1) the three different breastfeeding methods (i.e., only baby food, EB, integrated/combined breastfeeding) and (2) breastfeeding duration, chi-square tests were performed. No statistically significant differences were found among the educational levels of the groups in relation to breastfeeding methods: χ^2^ (2, N = 274) = 4.01, *p* > 0.05, *r_c_* = 0.12, and no statistically significant differences were found among the educational levels of the groups in relation to breastfeeding duration: χ^2^ (2, N = 274) = 2.49, *p* > 0.05, *r_c_* = 10. However, the effect of education may be more pronounced during the initiation phase, and less so in the continuation phase. Other factors such as family support and employment status might play a larger role in the continuation of EB over time.

Although higher education is often linked to a greater awareness of the benefits of breastfeeding, it may also be associated with greater workforce participation. This could lead to earlier breastfeeding cessation due to time constraints and work-related challenges.

Non-academic mothers, despite a lower formal education, may receive stronger familial encouragement to continue EB, counteracting the expected influence of education.

Cultural and Community Influences: In Arab society, breastfeeding decisions may be less influenced by formal education and more by cultural norms, family traditions, and religious beliefs. Since Arab families, especially in rural areas, strongly support breastfeeding regardless of education, the expected effect of education on breastfeeding duration may be diluted.

In order to test hypothesis H5, the assessment of religiosity level differences between (1) the three different breastfeeding methods (i.e., only baby food, EB, integrated/combined breastfeeding) and (2) breastfeeding duration, one-way analyses of variance (one-way ANOVAs) were employed.

In gauging the association between religiosity level and breastfeeding methods (exclusive; N = 65, M = 4.70, SD = 0.65/combined; N = 66, M = 4.37, SD = 0.71/baby food; N = 43, M = 4.27, SD = 0.97), the results indicate a significant main effect—F(2, 271) = 8.70, *p* < 0.01. Hence, in order to reveal the source of these significant differences, Tukey’s HSD post hoc tests were performed, finding that those who preferred EB reported higher religiosity levels than (1) those who preferred combined breastfeeding (*p* < 0.01) and (2) only baby food (*p* < 0.01). No other significant differences were discovered between these groups.

In gauging the association between religiosity level and breastfeeding duration during the last 4 months (did not breastfeed at all; N = 19, M = 3.97, SD = 1.24/up to 1 month; N = 10, M = 4.27, SD = 1.05/up to 2 months; N = 6, M = 3.61, SD = 0.83/up to 3 months; N = 25, M = 4.55, SD = 0.48/up to 4 months; N = 20, M = 4.35, SD = 0.90/above 4 months; N = 194, M = 4.67, SD = 0.61), the results indicate a significant main effect: *F*(5, 268) = 6.54, *p* < 0.01. In order to reveal the source of these significant differences, Tukey’s HSD post hoc tests were performed, indicating only that those who breastfed for >4 months reported a higher religiosity level than (1) those who preferred not to breastfeed (*p* < 0.01) and (2) those who breastfed up to 2 months (*p* < 0.01). No other significant differences were discovered between these groups.

In order to test hypothesis H6, the assessment of the association between prior instruction(s) received regarding breastfeeding (did not receive any/group instructional program/personal or private instructional program) and (1) the three different breastfeeding methods (i.e., only baby food, EB, integrated/combined breastfeeding) and (2) breastfeeding duration, chi-square tests were performed. No statistically significant differences were found among the instructional groups in relation to breastfeeding methods: χ^2^ (4, N = 274) = 4.28, *p* > 0.05, *r_c_* = 0.09. In addition, no statistically significant differences were found among the instructional groups in relation to breastfeeding duration: χ^2^ (10, N = 274) = 7.90, *p* > 0.05, *r_c_* = 0.12.

In order to test hypothesis H7, the assessment of the association between employment status (unemployed/part- or full-time salaried employee/self-employed) and (1) the three different breastfeeding methods (i.e., only baby food, EB, integrated/combined breastfeeding), chi-square tests were performed. No statistically significant differences were found among the different breastfeeding methods: χ^2^ (4, N = 274) = 17.08, *p* < 0.01, *r_c_* = 0.18. The results indicate that (1) those who were unemployed preferred EB; (2) those who were self-employed preferred to opt for baby food only or combined breastfeeding; and (3) salaried employees preferred combined breastfeeding. However, in testing whether these results changed under different employment/job percentage (0% or unemployed/part-time as 25%/50%/75%/full-time with 100% employment), it was discovered that the results (previous paragraph) did not change at all based on the job scope, and stayed relatively similar across all the work percentage groups, respectively: (1) χ^2^ = 4.57, *p* > 0.05, for the unemployed group; (2) χ^2^ = 4.02, *p* > 0.05, for the 25% part-timers; (3) χ^2^ = 0.49, *p* > 0.05, for the 50% part-timers; (4) χ^2^ = 6.35, *p* > 0.05, for the 75% part-timers; and (5) χ^2^ = 1.96, *p* > 0.05, for the full-timers.

Lastly, in order to test hypothesis H8, the assessment of the differences between educational levels (non-academic/academic) on (1) the reasons encouraging the mothers to breastfeed their infant and (2) the reasons leading them to stop breastfeeding, descriptive statistics, independent-samples *t*-tests, and chi-square tests were performed. Notably, the nine reasons fostering the mothers to breastfeed their infant were rated on a 3-point Likert-type scale (0 = did not influence my decision to breastfeed my baby; 1 = somewhat influenced my decision; 2 = influenced my decision a great deal). Moreover, the six reasons leading to a halt in breastfeeding were calculated on a dichotomous scale (0 = was not a reason to stop breastfeeding; 1 = was a reason to stop breastfeeding). The 3-point Likert scale was chosen for its simplicity and clarity, and to minimize participant confusion and response fatigue, thus enabling the clear categorization of levels of influence,

Depicted in Figure 1 are the reasons, in descending order, based on the mean scores, favoring breastfeeding. As can be seen, the two most prominent participant-reported reasons favoring breastfeeding were (1) wanting to maintain the baby’s health (M = 1.87) and (2) wanting to develop a more intimate relationship/attachment with the baby (M = 1.84).

Portrayed in Figure 2 are the participant-reported reasons leading to a halt in breastfeeding in a descending order based on relative frequencies. “Reported reasons” was used to describe the participants’ self-identified reasons for ceasing breastfeeding, as collected through the survey instrument. This phrasing reflects the subjective nature of the responses provided by the participants. As can be seen, the most prominent reported reason for halting breastfeeding was the quality of the milk/the amount of milk produced by the mother herself (50.7%).

Independent sample *t*-tests revealed several significant differences between academics and non-academics in terms of fostering breastfeeding. The results are presented in Table 8. As can be learned from the statistically significant effects, (1) non-academics were more influenced by their family members to breastfeed than academics, (2) academics were more influenced by instructional training programs than non-academics, (3) non-academics were more influenced by successful previous breastfeeding than academics, (4) non-academics were more influenced by financial constraints (i.e., EB does not require ‘special’ funds) than academics, and (5) academics tended to be more influenced by information obtained from the literature, internet, or media.

The chi-square tests indicated several significant associations (between educational level and reasons leading to a halt in breastfeeding). The chi-square test was found to be significant (χ^2^ = 6.21, *p* = 0.013). The results of the table show that 59.6% of the non-academics and 55.6% of the academics, indicated that the quality of the milk/the amount of milk produced by the mother herself was found to be a factor that influenced the decision to stop breastfeeding. (2) The chi-square test was found to be significant (χ^2^ = 11.75, *p* = 0.001). The results of the table show that 59.6% of the non-academics and 78.8% of the academics did not stop breastfeeding for the reason mentioned - the baby’s health issues. (3) No significant association was found with regard to the health issues of the mother as a reason between educational groups (χ^2^ = 1.06, *p* = 0.303). (4) The chi-square test was found to be significant (χ^2^ = 4.26, *p* = 0.039). The results of the table show that 84.2% of the non- academics and 73.8% of the academics did not stop breastfeeding for the reason mentioned—fatigue/inconvenience and lack of time or dissatisfaction from the breastfeeding experience. (5) No significant association was found with regard to the lack of support from professionals to continue breastfeeding as a reason between educational groups (χ^2^ = 2.19, *p* = 0.139). (6) The chi-square test was found to be significant (χ^2^ = 3.59, *p* = 0.051). The results of the table show that 80.7% of the non-academics and 70.6% of the academics did not stop breastfeeding for the reason mentioned—return to work and unsupportive conditions for breastfeeding an infant (Table 9).

## 4. Discussion

EB offers significant health benefits for both infants and mothers, applicable in both the short and long term. Breast milk provides essential nutrition and protection against infections and diseases [41], containing antibodies, cytokines, and antimicrobial compounds that support the infant’s immune system and facilitate its development. Breastfeeding reduces the risk of SIDS, with the risk being even lower with EB. Furthermore, EB protectively affects the development of asthma, eczema, type 1 diabetes, and other diseases. Infants who are breastfed have a lower risk of obesity. Furthermore, studies have shown that women who breastfeed experience lower incidence rates of breast and ovarian cancer [42,43,44].

In 2003, the WHO recommended EB for infants up to 6 months of age [43,44]. In Israel, a survey showed that the breastfeeding initiation rate is ~90%, with all women who began breastfeeding in the hospital reporting practicing EB. However, EB rates dropped significantly during maternity leave, largely due to challenges such as milk supply issues and technical difficulties with breastfeeding [45].

According to data from the 2nd National Health and Nutrition Survey, the percentage of women practicing EB at 4 to 6 months postpartum was only 28.8% (National Program for Nutrition and Health Surveys, 2020; Ministry of Health, 2023) [19,46]. Following these findings, a quality indicator was established within Israel’s National Program for Quality Indicators in Well-Baby Clinics (“Tipat Chalav”), focusing on “the rate of women practicing EB up to 4 months.” Data from 2022 show that the national rate for EB up to 4 months was 22%; whereas, the rate among Arab Israeli women was even lower, 14% [30]. These findings highlight Arab Israeli women as a group at an increased risk of lower rates of EB. This population’s specific characteristics and challenges should be investigated to better understand and address the factors affecting EB within this demographic, in line with the quality index’s goal of improving early nutrition and breastfeeding rates across Israel.

The largest Arab population in Israel is concentrated in the Northern District’s towns and villages. Arab citizens of Israel, who are primarily Muslim, Christian, and Druze, make up 21% of the overall population of Israel (Central Bureau of Statistics, 2020) [47]. The current study emphasizes the complex effects of religious beliefs, socio-demographic factors, and cultural norms regarding breastfeeding patterns among Arab mothers in Israel. These findings provide insights into the key factors shaping breastfeeding behavior, including religiosity, education, age, number of children, employment, and family support.

To elaborate, one of the central findings of this study is the strong association between high levels of religiosity and longer durations of EB. This finding aligns with the religious traditions in Islam, Christianity, and the Druze communities, which advocate for breastfeeding and regard it as the mother’s health-promoting and spiritual obligation towards the child. The religious value attributed to breastfeeding serves as a significant motivator for its continuation, particularly in Muslim and Druze communities where faith plays a central role in daily life. This finding aligns with Alchalel et al. [30], who reported that religious beliefs in these communities frame breastfeeding as both a maternal duty and a religious obligation. In contrast to Mehrpisheh et al. [36], who found that religious influence on breastfeeding was more pronounced amongst Christian mothers in other Middle Eastern countries, our study found that Christian Arab mothers in Israel exhibited lower EB rates. While all three religions value breastfeeding, Muslim mothers may follow practical adaptations, such as using a wet nurse (milk kinship) when direct breastfeeding is not possible. In contrast, Christian and Druze communities emphasize the mother–infant bond throughout prolonged breastfeeding, potentially influencing higher EB rates [30,31]. 

As the mother’s age increases, the likelihood of EB and its duration increases. Based on the data analyzed herein, we found that the relationship between the mother’s age and breastfeeding was significant. Older mothers tended to practice EB more often and for longer periods of time than younger mothers, as evidenced by the ANOVA findings showing significant differences in breastfeeding methods and duration between the different age groups. The post hoc analysis further confirmed that mothers who practiced EB were slightly older than those who used combined feeding methods, thus suggesting that older mothers may have more experience, patience, or access to resources that encourage EB. Moreover, as the mother has had more previous births (i.e., more children), the likelihood of EB and its duration increases.

The chi-square tests revealed statistically significant differences in breastfeeding practices based on the number of children a mother had birthed before her recent delivery. Mothers with more children, particularly those with three or more children, showed a stronger preference for EB compared to mothers with fewer or no prior children. This could be attributed to increased confidence and experience in breastfeeding as a mother has more children, as well as greater familiarity with overcoming initial breastfeeding challenges.

Furthermore, differences were found relating to the rate of EB and the duration of breastfeeding based on belonging to an ethnic group and/or geographic region. Ethnic and regional differences were observed to statistically and significantly impact both breastfeeding methods and duration. Muslim Arab mothers were more likely to combine breastfeeding with baby formula, whereas Christian and Druze mothers favored EB. Geographically, mothers from northern Israel were more likely to practice EB compared to those from the central or southern regions.

The majority of Muslim mothers in our sample resided in urban or mixed cities, where access to formula and alternative feeding options may have been more readily available. Druze mothers, who mostly lived in villages, may have had stronger familial and community support that encouraged sustained breastfeeding [30,31]. 

In some Muslim communities, there is a stronger reliance on combined breastfeeding (breastfeeding plus formula), possibly influenced by an early return to work, urbanization, and modern lifestyle changes. Druze women may adhere to more traditional family structures that emphasize maternal roles and longer breastfeeding durations. Amongst Christian Arab women, education and a higher socio-economic status may play a role in a greater awareness of the benefits of EB, leading to higher breastfeeding initiation and duration rates [30,31]. 

Moreover, as the educational level of the mother increases, the probability of EB and its duration increase. Contrary to expectations, the relationship between education level and breastfeeding was not as strong as hypothesized. The chi-square tests indicated no statistically significant differences between education levels and breastfeeding methods or duration. However, the effect of education may be more pronounced during the initiation phase, and less so in the continuation phase. Other factors such as family support and employment status might play a larger role in the continuation of EB over time.

Cultural and Community Influences: In Arab society, breastfeeding decisions may be less influenced by formal education and more by cultural norms, family traditions, and religious beliefs. Since Arab families, especially in rural areas, strongly support breastfeeding regardless of education, the expected effect of education on breastfeeding duration may be diluted.

Furthermore, as the mother’s level of religiosity increases, the likelihood of EB and its duration increase. The study found a significant association between higher religiosity and EB. Mothers with higher levels of religiosity were more likely to practice EB and for longer durations. In Islam, breastfeeding is considered a religious duty, with the Quran recommending a breastfeeding duration of two years [33]. Islam also acknowledges “milk kinship,” further emphasizing its spiritual significance [48]. In Christianity, breastfeeding is seen as a maternal duty, symbolizing love, care, and nourishment, with religious imagery often portraying the Virgin Mary breastfeeding Jesus. The Druze community also holds breastfeeding in high regard, viewing it as an essential part of motherhood.

In religious communities, extended family members (especially grandmothers) often encourage breastfeeding, reinforcing it as a cultural expectation. Community norms within religious societies may discourage formula feeding, associating breastfeeding with traditional maternal roles [32]. Moreover, many religious mothers, particularly in traditional Muslim and Druze communities, do not work outside the home or take extended maternity leave, which facilitates longer breastfeeding durations. Employment is negatively correlated with EB, as working mothers often switch to formula feeding earlier due to workplace constraints. This reflects the religious beliefs in these communities that encourage breastfeeding as a spiritual and health-related obligation and underscores the importance of integrating religious considerations into public health strategies to promote breastfeeding.

However, as opposed to our hypothesis, no statistically significant differences were found as to the type of breastfeeding training mothers received and their breastfeeding methods or duration. Whereas breastfeeding courses and training may provide initial support, they did not seem to affect long-term breastfeeding behavior in this sample, hence suggesting that while training may help with initiation, other factors such as family support, personal experience, and cultural norms may play a more substantial role in sustaining EB.

We initially hypothesized that participation in breastfeeding courses would increase EB duration, yet this was not supported. A possible explanation is that such courses focus more on initiation rather than long-term breastfeeding strategies. Additionally, mothers may rely more on family advice and social support than formal training when making breastfeeding decisions.

In line with our other hypotheses, working reduced the likelihood of both EB and its duration, further decreasing as workload (volume) increased. The results indicate that employment status significantly affected breastfeeding practices. Unemployed mothers were more likely to practice EB; whereas, mothers who were employed, especially, those with full-time jobs, tended to opt for combined feeding methods. Employment-related factors such as a lack of time and unsupportive workplace conditions contributed to early cessation of breastfeeding. These findings highlight the need for workplace policies that support breastfeeding, such as extended maternity leave and on-site breastfeeding facilities.

While we found that employment negatively affected breastfeeding, the degree of employment (full-time vs. part-time) did not significantly alter breastfeeding duration which might suggest that even part-time employment creates barriers to long-term EB, particularly in environments with limited workplace breastfeeding accommodations. This is consistent with Zimmerman et al. (2022) [45], who found that maternity leave policies and workplace support play a crucial role in breastfeeding continuation. However, Carandang et al. [49] found that in Western settings, higher maternal education is often linked to longer EB durations, whereas our study found no significant relationship between academic level and breastfeeding duration, thus suggesting cultural or socio-economic variations in how employment and education intersect with breastfeeding behaviors.

Similarly to Arif et al. [30], our study found that milk insufficiency, fatigue, and work-related constraints were the most cited reasons for stopping breastfeeding. However, the Western literature [50] emphasizes a lack of professional support as a major barrier to breastfeeding, whereas our study found that professional breastfeeding training had no significant impact on EB duration.

To summarize our findings -religious beliefs played a significant role in breastfeeding duration, particularly, amongst Muslim and Druze women, who often view breastfeeding as both a religious and health obligation. Religious mothers tended to breastfeed longer.

No significant link was found between education level and breastfeeding duration. However, academic mothers were more likely to be influenced by the medical literature, while non-academic mothers were influenced by family and financial factors. Muslim Arab mothers were more likely to use both breastfeeding and formula feeding, while Christian and Druze mothers preferred EB. Regionally, mothers in northern Israel were more likely to practice EB than those in central or southern Israel, reflecting regional cultural norms.

Employment significantly affected breastfeeding practices. Unemployed mothers were more likely to practice EB; whereas, full-time working mothers tended to use a combination of breastfeeding and formula. Workload and unsupportive workplace policies contributed to early breastfeeding cessation.

Family support, especially from grandmothers and older women, played a crucial role in encouraging breastfeeding, particularly in Arab communities. Including family members in breastfeeding education could improve EB rates.

Common reasons for stopping breastfeeding included insufficient milk supply, fatigue, and a return to work. Concerns about milk supply and baby health affected non-academic mothers more while academic mothers cited work-related issues, a lack of time, and fatigue.

Surprisingly, breastfeeding training did not significantly affect breastfeeding duration. More influential factors were religious beliefs, family support, and employment status. Cultural and social factors were found to play a more critical role than education or training in determining breastfeeding practices.

In conclusion, this study has provided evidence supporting most of the hypotheses, particularly, those related to age, number of children, religiosity, and employment status. Yet, the effect of educational level and breastfeeding training on breastfeeding practices was less pronounced than expected.

### 4.1. Policy Recommendations

Our findings emphasize the need to integrate religious and cultural factors into breastfeeding promotion programs within religious communities. Healthcare professionals should be aware of the religious and cultural significance of breastfeeding and provide support to mothers in alignment with their beliefs. As such, encouraging EB should commence early in the pregnancy, during hospitalization, and continue after discharge within the community. Furthermore, raising awareness among mothers and their surroundings will aid in adopting health-promoting behaviors through breastfeeding and prevent its early discontinuation.

Collaborative efforts should be established amongst all members of the healthcare team (physicians, nurses, dietitians, and social workers) that offer lactation support by:Establishing community-based breastfeeding support programs, integrating religious leaders and family elders to align with cultural preferences.Increasing government funding for lactation consultants and integrating breastfeeding counseling into prenatal and postnatal care visits.Developing mobile lactation units to reach rural or underserved communities, ensuring equitable access to breastfeeding resources.

Workplace policies should be enacted that support breastfeeding by the following means:Implement paid maternity leave of at least six months, as per WHO and UNICEF recommendations.On-site lactation rooms with comfortable seating, refrigeration, and privacy.Flexible work hours and remote work options for breastfeeding mothers.Mandatory breastfeeding breaks integrated into work schedules.Government incentives (e.g., tax reductions or subsidies) to businesses that adopt breastfeeding-friendly policies.

Integrate breastfeeding practice using religious and cultural norms with public health campaigns, as follows:Partner with religious institutions to promote breastfeeding, using media campaigns and local success stories.Create mentorship programs connecting experienced mothers with new ones.Incorporate breastfeeding education in schools and universities, mandate training for healthcare staff, and ensure hospitals adopt the Baby-Friendly Hospital Initiative to support breastfeeding from birth.

Legislative action can also be taken to enforce anti-discrimination laws protecting breastfeeding mothers in workplaces and public spaces; penalize violations of maternity leave rights to prevent premature return pressures; and subsidize breast pumps and breastfeeding supplies for low-income families. By integrating workplace policies, healthcare initiatives, cultural adaptation strategies, and legal protections, breastfeeding rates amongst Arab Israeli mothers can be significantly improved. These targeted actions will help bridge socio-economic gaps and ensure sustained support for mothers in both professional and community settings.

### 4.2. Limitations and Future Research Recommendations

The sampling method employed in the study utilized a snowball sampling technique; participants recruited additional participants from their personal networks.

We recommend that future studies employ randomized or stratified sampling methods to ensure a more representative sample. Alternatively, combining snowball sampling with other probability-based sampling techniques could reduce bias and increase generalizability.

The geographic limitations of the sample were noted, as most participants resided in northern Israel, which could affect the generalizability of the findings to other populations in different geographic areas of the country. We acknowledge that the findings may not fully represent Arab mothers from central and southern Israel, where socio-economic conditions and cultural influences may differ. To mitigate such biases, the survey was distributed across all residential areas of the Arab population. Fortunately, it achieved a satisfactory representation from all segments of the population, reflecting the diversity of the Arab community in Israel in terms of origin and place of residence.

We recommend that future studies employ a stratified or randomized sampling method to ensure a more balanced geographic representation. We also suggest cross-regional comparisons in future research to better understand potential differences in breastfeeding behaviors between northern, central, and southern Arab communities. Since participants were asked to retrospectively report on breastfeeding after childbirth, there is a likelihood of memory bias that may affect the accuracy of the reports, particularly, regarding the duration of breastfeeding and the reasons for its cessation. We recommend that to reduce memory bias, future research could prospectively collect data by following participants from childbirth through the breastfeeding period. Alternatively, researchers could use mixed methods, incorporating qualitative interviews to validate quantitative reports. The study relied on self-reported questionnaires, which may introduce social desirability bias - participants might have provided responses that align with social norms rather than their actual experiences. Additionally, responses regarding religiosity and breastfeeding duration could have been influenced by cultural expectations. Future research should incorporate objective measures, such as medical records or lactation consultant assessments, to validate self-reported data.

Furthermore, the study focused on Arab mothers in Israel; therefore, the findings may not be applicable or representative of other populations, even within Israel or in the broader global context, thereby, limiting the generalizability of the findings to other populations. We recommend conducting similar studies (i.e., systematic replications) with diverse ethnic and religious groups, both within Israel and in other countries, to determine whether the findings are culturally specific or universally applicable. Cross-cultural comparisons could also be valuable.

From a cultural/religious aspect, although the study emphasized religious and cultural beliefs, we did not explore in depth the interaction between these and other factors, such as modern social norms and global effects, which may alter breastfeeding patterns in specific populations. We recommend that future research explore the dynamic relationship between traditional cultural/religious beliefs and factors, such as media, globalization, and socioeconomic factors, to understand how these affect breastfeeding practices. This can be achieved through longitudinal studies and in-depth qualitative research.

## 5. Conclusions

We examined the impact of religious beliefs and socio-demographic factors on breastfeeding patterns among Arab mothers in Israel, focusing on Muslim, Christian, and Druze populations. We explored how religion, age, education, number of children, and employment impact the choice of exclusive or combined breastfeeding, finding a direct correlation between high levels of religiosity and longer durations of EB, particularly, among Druze and Muslim women. Additional factors affecting breastfeeding duration include family support, especially from older women in the family, as well as economic and employment-related factors. Working mothers were more likely to combine breastfeeding with infant formula.

The research revealed that the motivations for continued breastfeeding were primarily centered around child health and the intimate bond between mother and child. Conversely, a lack of breast milk was the most common reason cited for ceasing breastfeeding. We underscore the need for broader, multidisciplinary support from the healthcare system and employers for breastfeeding mothers, as well as programs tailored to the religious and cultural backgrounds of the studied populations.

## Figures and Tables

**Figure 1 healthcare-13-00852-f001:**
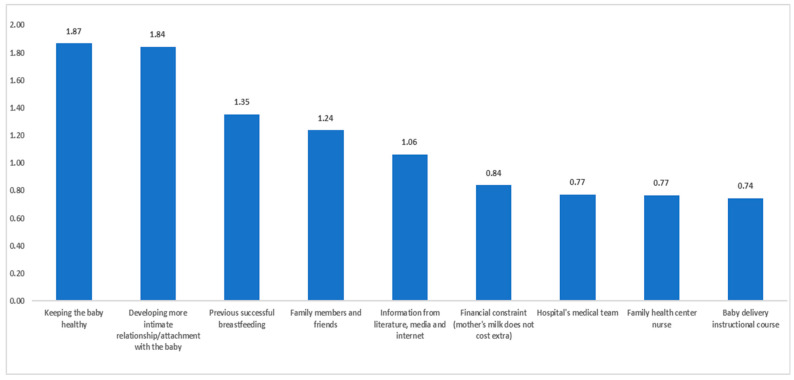
Reasons supporting breastfeeding (mean scores).

**Figure 2 healthcare-13-00852-f002:**
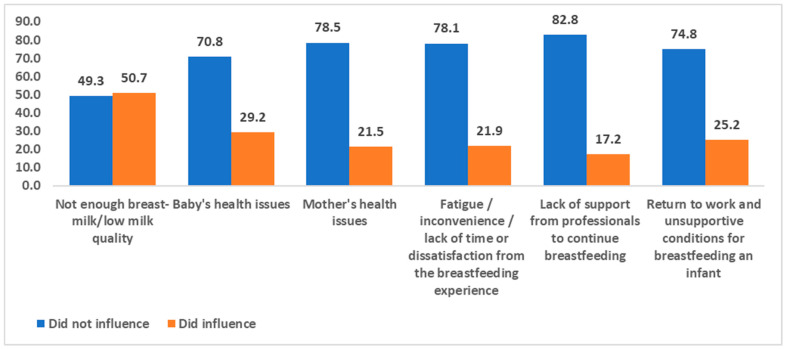
Participant-reported reasons for halting breastfeeding with relative frequencies (proportions).

**Table 1 healthcare-13-00852-t001:** Participant demographics (N = 274).

Characteristic	Category	n (%)
Age (years)	Mean (SD) = 31.05 (5.21)	
Religious Affiliation	Muslim	186 (67.9%)
	Christian	40 (14.6%)
	Druze	48 (17.5%)
Geographic Region	North	220 (80.3%)
	Center	19 (6.9%)
	South	35 (12.8%)
Education Level	Did not finish high school	27 (9.9%)
	Completed high school	64 (23.4%)
	Academic degree	160 (58.4%)
Employment Status	Employed (full/part-time)	176 (64.2%)
	Self-employed	17 (6.2%)
	Unemployed	81 (29.6%)
Prior Number of Children	None	105 (38.3%)
	One	53 (19.3%)
	Two	48 (17.5%)
	Three or more	68 (24.8%)
Planned Return to Work	Yes	102 (37.2%)
	No	24 (8.8%)
	Already returned	78 (28.5%)
	Undecided	70 (25.5%)

**Table 2 healthcare-13-00852-t002:** H2—number of previous children and breastfeeding duration.

	Breastfeeding Duration						
	Did Not Breastfeed at All	m	2 m	3 m	4 m	>4 m	Total
No	14	10	5	10	10	56	105
% within	13.3%	9.5%	4.8%	9.5%	9.5%	53.3%	100%
1 child	3	0	1	6	5	38	53
% within	5.7%	0%	1.9%	11.3%	9.4%	71.7%	100%
2 children	2	0	0	4	3	39	48
% within	4.2%	0%	0%	8.3%	6.3%	81.3%	100%
3 children	0	0	0	5	2	61	68
% within	0%	0%	0%	7.4%	2.9%	89.7%	100%
Total	19	10	6	25	20	194	274
% within	6.9%	3.6%	2.2%	9.1%	7.3%	70.8%	100%

**Table 3 healthcare-13-00852-t003:** H2—number of children prior to recent delivery and breastfeeding methods.

	Breastfeeding Methods			Total
	Only Baby Food	Integrated/Combined	EB	
No children	30	31	44	105
% within	28.6%	29.5%	41.9%	100%
1 child	7	14	32	53
% within	13.2%	26.4%	60.4%	100%
2 children	4	14	30	48
% within	8.3%	29.2%	62.5%	100%
3 children	2	7	59	68
% within	2.9%	10.3%	86.8%	100%
Total	43	66	165	274
% within	15.7%	24.1%	60.2%	100%

**Table 4 healthcare-13-00852-t004:** H3-differences in exclusive breastfeeding rate and duration by ethnic group and geographical region.

	Breastfeeding Methods			Total
	Only Baby Food	Integrated/Combined	EB	
Muslim	38	55	93	186
% within	20.4%	29.6%	50%	100%
Christian	5	8	27	40
% within	12.5%	20%	67.5%	100%
Druze	0	3	45	48
% within	0%	6.3%	93.8%	100%
Total	43	66	165	274
% within	15.7%	24.1%	60.2%	100%

**Table 5 healthcare-13-00852-t005:** H3—breastfeeding duration across ethnic groups.

	Breastfeeding Duration						
	Did Not Breastfeed at All	>m	>2 m	>3 m	>4 m	<4 m	Total
Muslim	17	8	4	22	17	118	186
% within	9.1%	4.3%	2.2%	11.8%	9.1%	63.4%	100%
Christian	2	1	2	3	2	30	40
% within	5%	2.5%	5%	7.5%	5%	75%	100%
Druze	0	1	0	0	1	46	48
% within	0%	2.1%	0%	0%	2.1%	95.8%	100%
Total	19	10	6	25	20	194	274
% within	6.9%	3.6%	2.2%	9.1%	7.3%	70.8%	100%

**Table 6 healthcare-13-00852-t006:** H3—breastfeeding methods by geographic region.

	Breastfeeding Methods			Total
	Only Baby Food	Integrated/Combined	EB	
North	35	46	139	220
% within	15.9%	20.9%	63.2%	100%
Center	6	4	9	19
% within	31.6%	21.1%	47.4%	100%
South	2	16	17	35
% within	15.7%	24.1%	60.2%	100%
Total % within	43	66	165	274

**Table 7 healthcare-13-00852-t007:** H3—breastfeeding duration by geographic region.

	Breastfeeding Duration						
	Did Not Breastfeed at All	>m	>2 m	>3 m	>4 m	<4 m	Total
North	15	7	3	15	13	167	220
% within	6.8%	3.2%	1.4%	6.8%	5.9%	75.9%	100%
Center	3	1	0	3	2	10	19
% within	15.8%	5.3%	0%	15.8%	10.5%	52.6%	100%
South	1	2	3	7	5	17	35
% within	2.9%	5.7%	8.6%	20%	14.3%	48.6%	100%
Total	19	10	6	25	20	194	274
% within	6.9%	3.6%	2.2%	9.1%	7.3%	70.8%	100%

**Table 8 healthcare-13-00852-t008:** Independent sample t-test results, means, and standard deviations describing the factors that foster breastfeeding.

Reason	Group	N	M	SD	*t*-Test
Family members and friends	Non-academic	114	1.38	0.81	*t* = 2.26, *p* < 0.05
	Academic	160	1.14	0.90	
Family health center nurse	Non-academic	114	0.71	0.75	*t* = 0.98, *p* > 0.05
	Academic	160	0.81	0.83	
Hospital’s medical team	Non-academic	114	0.70	0.75	*t* = 1.28, *p* > 0.05
	Academic	160	0.83	0.81	
Baby delivery instructional course	Non-academic	114	0.56	0.76	*t* = 3.13, *p* < 0.01
	Academic	160	0.88	0.85	
Previous successful breastfeeding	Non-academic	114	1.59	0.77	*t* = 3.79, *p* < 0.01
	Academic	160	1.19	0.92	
Financial constraint (mother’s milk does not cost extra)	Non-academic	114	1.16	0.94	*t* = 5.00, *p* < 0.01
	Academic	160	0.61	0.85	
Developing a more intimate relationship or attachment with the baby	Non-academic	114	1.81	0.56	*t* = 1.00, *p* > 0.05
	Academic	160	1.87	0.45	
Keeping the baby healthy	Non-academic	114	1.85	0.50	*t* = 0.52, *p* > 0.05
	Academic	160	1.88	0.45	
Information from the literature, media, and internet	Non-academic	114	0.78	0.89	*t* = 4.58, *p* < 0.01
	Academic	160	1.26	0.84	

**Table 9 healthcare-13-00852-t009:** The significant chi-square tests between educational levels and the reasons leading to a halt in breastfeeding.

Reason	Group	Did Influence	Did Not Influence	Chi-Square Test
1. The quality of the milk/the amount of milk produced by the mother herself	Non-academic	59.6%	40.4%	χ^2^ = 6.21,
	Academic	44.4%	55.6%	*p* = 0.013
2. The baby’s health issues	Non-academic	40.4%	59.6%	χ^2^ = 11.75,
	Academic	21.3%	78.8%	*p* = 0.001
4. Fatigue/inconvenience and lack of time or dissatisfaction with the breastfeeding experience	Non-academic	15.8%	84.2%	χ^2^ = 4.26,
	Academic	26.3%	73.8%	*p* = 0.039
6. Return to work and unsupportive conditions for breastfeeding an infant	Non-academic	19.3%	80.7%	χ^2^ = 3.59,
	Academic	29.4%	70.6%	*p* = 0.051

## Data Availability

Individual-level data cannot be made publicly available due to legal and ethical restrictions. Aggregative data might be provided upon reasonable request to the corresponding author.

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
