# Peer review of "Factors Influencing Exclusive Breastfeeding Amongst Arab Mothers in Israel: Social, Cultural, and Occupational Aspects"

_healthcare, 2025, doi:10.3390/healthcare13080852_

Round 1
Reviewer 1 Report
Comments and Suggestions for Authors
Thank you for your hard work. I read this article with interest, but I have some concerns:
The Introduction should examine the literature in more depth. Existing studies should be discussed more comprehensively. In particular, more references should be provided to previous studies conducted among Arab mothers in Israel.
Why is it assumed that more religious mothers will breastfeed more? Are there similar findings in the literature?
The need for research questions is not emphasized. It should be clearly stated how this study fills a gap in the literature.
The Methods Section does not fully explain how the variables measured were analyzed. For example, how was the level of religious belief measured? Was a Likert scale used? The exact criteria should be stated.
There may be bias in the data collection process. Data were collected mostly from Arab communities in the north; therefore, it may not represent Arab communities in different regions of Israel.
The sample selection may be biased. Snowball sampling was used, but the biases introduced by this method are not sufficiently discussed.
The Results Section does not discuss possible reasons for the differences between groups. For example, why do Christian and Druze mothers have higher breastfeeding rates than Muslim mothers? The socioeconomic or cultural reasons for this should be explained.
Negative findings are not sufficiently discussed. Some hypotheses are not supported, but the reasons are not explained. For example, no significant difference was found between education level and breastfeeding duration (H4 hypothesis). Possible reasons for this should be discussed.
Findings should be summarized clearly. For example: "More religious mothers continued breastfeeding for longer periods, but no significant relationship was found between education level and breastfeeding duration."
In the Discussion Section, the comparison of the results with those of previous studies is weak. It should be explained more clearly how the findings agree or differ from the existing literature.
Limitations such as sampling bias, recall bias, and self-reported data are not sufficiently discussed.
The findings are not sufficiently related to the literature. Similarities and differences should be emphasized by referencing previous studies.
Policy recommendations in the Conclusion Section are vague: For example, it is stated that "policies supporting breastfeeding in the workplace should be developed," but no specific recommendations are provided.
Comments on the Quality of English Language
I do not consider myself competent in evaluating the Quality of English Language.
Reviewer 2 Report
Comments and Suggestions for Authors
This study examines the factors associated with breastfeeding among Arab mothers in Israel. While the introduction and discussion sections are well-written, the results section is primarily composed of text, making it very difficult for readers to interpret and understand the findings. Therefore, substantial revisions are necessary for the way authors present their analyses. Below are comments.
Since the study identifies mere correlations between factors and outcomes, causal language (e.g., influence, cause, effect, impact) should be avoided. Accordingly, the title should be revised (e.g., "Factors Associated with ~").
The statistical methods used for hypothesis testing (anova, x2) should be clarified in Section 2.5 (Data Analysis) rather than in the results section. Additionally, details regarding how the post-hoc analysis was conducted should also be included in this section.
The content presented in Section 2.2 (Participants) belongs in the results section rather than the methods section. Furthermore, a descriptive table should be added to present this information instead of solely relying on the text.
Section 2.4 (Procedure) should be positioned immediately before or after the section on ethical approval.
P-values should not be presented as "<0.05," ">0.05," or "<0.01" but should instead be reported with exact three-decimal values (e.g. p=0.110).
Line 302: "M=0.05" – Please correct this typographical error.
*****Although the chi-square test results are presented, the authors do not provide the counts for each cell, making interpretation challenging. A table displaying this information is essential to understanding the results for readers. How can readers infer correlations between variables and outcomes based solely on the chi-square test’s degrees of freedom and values?*****
For the same reason, results should not be presented only in text form; instead, tables should be provided to appropriately display the relevant information. This applies to all hypotheses tested in the results section.
Lines 443–456: This content is presented as a table footnote rather than as part of the main text, which warrants the correction. Additionally, when reporting chi-square test results, a corresponding table must be included.
Round 2
Reviewer 1 Report
Comments and Suggestions for Authors
I would like to thank the authors for their detailed and thoughtful responses to the previous major revision comments.
The revised manuscript shows significant improvement in both depth and clarity.
Given these comprehensive revisions and the overall contribution of the study to the field of public health and maternal-child health, I find the current version suitable for publication in Healthcare.
I recommend acceptance of the manuscript in its present form.